# Demographic, social and health system factors associated with maternal mortality in Pakistan: A nested case-control study

Ahsan Maqbool Ahmad[1]*, Iqbal H. Shah[2], Ali Muhammad Mir[3], Maqsood Sadiq[3], Muddassir Altaf Bosan[4]

1 PRIME Consulting, Islamabad, Pakistan, 2 Harvard T.H. Chan School of Public Health, Boston, MA, United States of America, 3 The Population Council, Islamabad, Pakistan, 4 Ministry of National Health Services, Regulations and Coordination, Islamabad, Pakistan

* drahsen@yahoo.com

## Abstract

**Data Availability Statement:** The data underlying the results presented in the study are available from https://dhsprogram.com/data/available-datasets.cfm

### Background

Pakistan has experienced a significant reduction in maternal mortality with a decline of 33 percent between 2006 and 2019. However, the country still grapples with a high number (186 per 100,000 live births) of maternal deaths each year. This study aims to identify socio-demographic and health system related factors associated with maternal mortality.

### Methods

Using the nested case-control design, we conducted an in-depth analysis of Pakistan Maternal Mortality Survey (PMMS) 2019. We identified 147 maternal deaths occurring within three years prior to the PMMS 2019 as "cases" and 724 women who gave birth and were alive during the same period as "controls". Socio-demographic characteristics of cases and controls were compared, and multivariate regression was employed to investigate the predictors of maternal mortality in Pakistan.

### Results

Cases and controls were similar on access to antenatal care (ANC) and ANC provider but differed on age, education, number of pregnancies, type of delivery, tetanus toxoid vaccination during last pregnancy, and contraceptive usage. A higher proportion of cases had deliveries by skilled birth attendants (83% compared to 63% among controls) while home deliveries were more common among controls (32% compared to 25% among cases). Odds of maternal death were lowest among women aged 20–29 years (odds ratio–OR: 0.5; 95% CI 0.23–1.07) and those with secondary or higher education (OR: 0.35; 95% CI 0.17–0.74). Surprisingly, deliveries attended by skilled birth attendants were associated with higher odds of maternal death (OR: 4.07; 95% CI 2.19–7.57) compared to those who were not.

**Funding:** The authors received no specific funding for this work.

**Competing interests:** The authors have declared that no competing interests exist.

## Conclusion

This study identifies secondary or higher maternal education, having had tetanus injection during the last pregnancy, ever-used contraception or being in the age group of 20–29 years were factors associated with lower risk of maternal mortality. Conversely, skilled birth attendance increases the risk of maternal death in Pakistan. Further investigation is needed into the determinants of high maternal mortality.

## Introduction

Childbirth is typically a joyful occasion, yet 287,000 (uncertainty interval–UI: 273,000 to 343,000) women were estimated to have died globally of maternal causes during pregnancy, childbirth or within 42 days of pregnancy termination in 2020 [1]. Although there has been 66.9 percent reduction in maternal mortality ratio (MMR) from 417 in 2000 to 138 maternal deaths per 100,000 livebirths in 2020, South Asia accounted for 16.4 percent of all maternal deaths in 2020 globally [1]. Notably Pakistan exhibited a higher MMR in 2020 at 154 compared to other Muslim majority countries such as Bangladesh (123), Iran (22) and Egypt (17) and other South Asian countries like India (103) and Sri Lanka (29) [1]. Pakistan has witnessed a decline in MMR from 276 in 2006–07 [2] to 186 (CI: 138–234) in 2019 [3], representing a one-third decline in the number of maternal deaths between 2007 and 2019. It is worth noting that these estimates pertain to a period of three years prior to each survey. However, given that the Sustainable Development Goal (SDG) target for reducing maternal mortality is to lower the MMR to less than 70 maternal deaths per 100,000 live births by 2030 [1], Pakistan still faces the imperative to make substantial progress on this indicator particularly in addressing the high burden of maternal mortality among some population subgroups.

Studies and surveys in Pakistan show estimates of MMRs ranging from 276 in 2006–07 and 279 in 2000–2001 to 401 in 2014 [2, 3, 9]. The diversity in estimates of MMR has been due to the use of different measurement approaches. Model-based estimates differ from those based on verbal autopsy data collected in cross-sectional surveys such as Demographic and Health Surveys (DHS). Apart from a study [8], very few have examined the predictors of maternal mortality, and none has applied a conceptual framework to guide the analysis.

Both Pakistan Demographic and Health Survey (PDHS) and Pakistan Maternal Mortality Survey (PMMS) provide valuable information on the level of maternal mortality nationally, by region and urban-rural place of residence. However, there remains a dearth of understanding regarding the contextual, health system, social, and biological factors that significantly influence the reduction of preventable maternal deaths in Pakistan. By identifying subgroups of women with high risk of maternal mortality, this study aims to call for concerted efforts focusing on these groups. The study also aims to pinpoint interventions that are needed to reduce preventable maternal deaths in Pakistan. In addition, it highlights the application of case-control methodology to the analysis of maternal mortality data collected in cross-sectional surveys such as DHS and Maternal Mortality Surveys.

## Materials and methods

Building upon prior research on the determinants of maternal mortality [4–8], we developed a conceptual framework to explore the predictors of maternal mortality using the 2019 Pakistan Maternal Mortality Survey (PMMS) data that used a verbal autopsy tool. Specifically, we

hypothesize that socioeconomic and demographic characteristics and health seeking behavior and comorbidities jointly or individually are associated with the risk of maternal deaths in Pakistan and disproportionately impact certain population subgroups (Fig 1).

## Study design

We applied case-control study design to data from PMMS. PMMS and other cross-sectional surveys covering maternal deaths often use verbal autopsy module to ascertain information about the deceased. Case-control design is an appropriate option to study the outcome of maternal mortality where those who died are classified as "cases" and those who survived as "controls". The purpose of classifying into these two groups on outcome, that is, maternal deaths in this study, is to investigate the underlying factors that distinguish the two groups.

## Study settings

The data were obtained from the PMMS that was the first exclusive survey on maternal health, morbidity, and mortality in Pakistan. It was carried out by the National Institute of Population Studies (NIPS) in 2019. Using a multi-stage cluster design, this nationally representative survey was conducted across four provinces (Balochistan, Khyber Pakhtunkhwa–KP, Punjab and

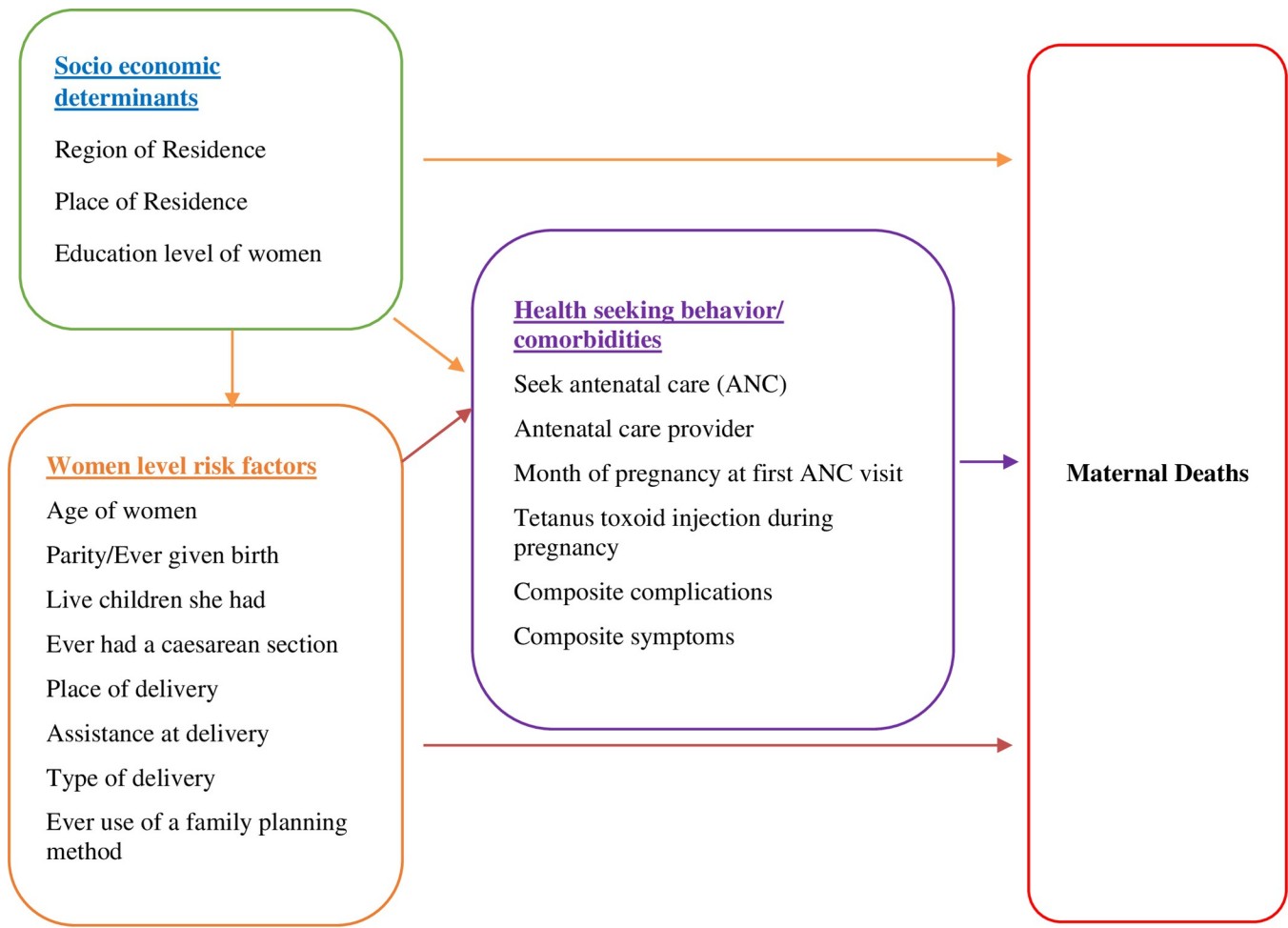

**Fig 1. Conceptual framework of predictors of maternal death.**

Sindh) as well as in Gilgit-Baltistan (GB) and Azad Jammu and Kashmir (AJ&K). These provinces and regions differ in terms of ethnicity and the main language spoken as well as literacy rate and access to health care facilities. The sampling design of 2019 PMMS provided estimates at the national level and for the four provinces (Balochistan, Khyber Pakhtunkhwa combined with Federally Administered Tribal Areas; Punjab combined with Islamabad Capital Territory; and Sindh) and two regions (GB and AJ&K). National estimates excluded GB and AJ&K.

## Data collection

Data collection for the 2019 PMMS was conducted in two phases. In the first phase, 11,859 ever-married women aged 15–49 in 108,766 households were interviewed. Among other questions, women were asked about births and deaths, including deaths among ever-married women aged 15–49 during the three years prior to the survey. Detailed verbal autopsies were conducted among households that reported at least one death of a woman aged 15–49 years. In the second phase, a subsample of households was randomly selected to provide information on women aged 15–49 including a complete pregnancy history.

We performed a nested case-control study in which all direct and indirect maternal deaths (147) identified in the PMMS were regarded as cases, while the controls were randomly selected by matching on the cluster from the rest of (6,907) women who reported a live birth during the last three years before the survey. The PMMS Household Questionnaire collected information on all deaths in the three years before the survey. All female deaths (1,177) were further investigated in detail using the verbal autopsy (VA) questionnaire, which was administered by specially trained interviewers. A total of 1,117 verbal autopsies were reviewed by a panel of nosologists (expert obstetricians and gynecologists and physicians) at the National Committee of Maternal and Neonatal Health (NCMNH). The panel classified maternal deaths into direct, indirect, coincidental, and late maternal deaths, based on the cause of death [9]. International Classification of Diseases 11[th] Revision (ICD-11) defines maternal death as the death of a woman while pregnant or within 42 days of termination of pregnancy, irrespective of the duration and site of the pregnancy, from any cause related to or aggravated by the pregnancy or its management but not from unintentional or incidental causes [10]. Direct maternal deaths are those "resulting from obstetric complications of the pregnant state (pregnancy, labor and puerperium), and from interventions, omissions, incorrect treatment, or from a chain of events resulting from any of the above" [10]. Indirect maternal deaths are those "resulting from previous existing disease or disease that developed during pregnancy, and that were not due to direct obstetric causes but were aggravated by the physiologic effects of pregnancy" [10]. Our study analyzes direct and indirect maternal deaths only.

The nested case-control design extends the flexibility of focusing specifically on those who experienced mortality (cases) and a set of matched controls per case providing the statistical strength to conduct analysis through which variability due to known and unknown risk factors related to various characteristics and health-seeking behaviors can be controlled appropriately.

There was a 1:5 ratio between cases and controls. All deaths of women of reproductive age that occurred during the last three years preceding the survey and classified as direct or indirect maternal deaths were included as "cases" in the study (N = 147). On the other hand, all women who were alive and reported live births during the same period and matched on the same clusters as cases were classified as "controls. The controls (N = 724) having the required information about risk factors were included for the analysis (Fig 2). To maximize the statistical power of analysis, inclusivity for statistical analysis purposes was considered in a way, so that information on maximum number of cases could be utilized. In this milieu, there were 136 cases that were included in the final analysis on the basis that they had at least one cluster matched control

**Fig 2. Scheme of cases and controls identification and matching.**

available. As a result, there were only four clusters for whom a lesser number of controls per case (i.e., less than ratio of 1:5) were available. The sample sizes were computed at the 95% confidence level, at a power of 90%, an assumed 20% rate of exposure among controls. Note that the information for cases (deceased women) was provided by the household member(s) most knowledge-able about the deceased woman's symptoms preceding her death and her background characteristics, whereas information about the controls was provided by women themselves.

### Study variables

Operationalizing the conceptual framework (shown in Fig 1), maternal death was taken as an outcome variable that was dichotomized into whether the death occurred or not (i.e.,

cases were maternal deaths, and controls were women who were alive and had a live birth). The proximate determinants of the outcome of maternal death are the health seeking behaviors such as seeking antenatal care (ANC); antenatal care provider; month of pregnancy at first ANC visit; receiving tetanus toxoid (TT) injection during pregnancies and comorbidities such as composite index of complications and composite symptoms. Socio-economic variables examined for their potential role in women level risk factors and health seeking behaviors to exacerbate or reduce the risk of maternal death are region and urban/rural place of residence and educational level of women. Women level risk factors include age; parity; number of living children; ever had caesarean section; place of delivery; assistance with delivery; type of delivery and contraceptive usage. Together, these variables attenuate or escalate the risk of maternal mortality.

## Statistical analysis and software

Multi-stage analyses included the phases of descriptive analysis and inferential analyses by applying univariate and multivariate regression techniques. The proportional representation of cases versus controls was similar for the matching variables of region/province of residence and place of residence (i.e., urban vs rural).

Frequencies and percentages were reported for categorical variables and mean with standard deviation for continuous variables (for example, composite complications and composite symptoms). Cross tabulations were developed for categorical variables to identify any sparse data necessitating merging with other categories.

Across the initial stages of the analyses, three sets of iterations were performed to re-define categories of independent variables. Given the small number of cases in the survey, the number of categories were merged for variables to allow for meaningful comparisons.

To determine the risk factors associated with maternal deaths, these factors were examined separately in univariate analysis. Crude odds ratios (ORs) and matched odds ratios (matched ORs) with 95% confidence intervals (CI) were computed by simple logistic regression and conditional logistic regression, respectively. Variables with a p-value less than 0.25 at the univariate level were included in the multivariable analysis and multi-collinearity among the independent variables was checked. Multivariable analysis was done using the same regression techniques to determine risk factors associated with maternal deaths by computing adjusted odds ratios (adjusted ORs) with 95% CIs. Variables having p-value less than 0.05 were kept in the final multivariable model and the Hosmer Lemeshaw goodness of fit test was applied to check goodness of fit of the model using simple logistic regression technique.

We assessed the impact of each of the specified risk factors on maternal mortality by estimating odds ratios (ORs) using a logistic regression model to control for the effects of known biological and socioeconomic variables (as shown in Fig 1). Analysis was done using Stata version 14 (a software for data analysis and management StatCorp, College Station, Texas).

## Ethical statement

We analyzed anonymized data available in the public domain for secondary analysis. The National Institute for Population Studies that collected the primary data observed all the required ethical considerations, including obtaining the requisite approvals and consent from respondents for the interview.

## Results

### Profile of cases and controls and risk factors

Table 1 shows the number and percentage distribution (%) of cases and controls by background variables. Similar proportion of cases and controls who had a live birth, stillbirth, miscarriage, or abortion in three years prior to the survey accessed antenatal care (ANC); and saw an obstetrician/gynecologist or a doctor for ANC. Cases were, however, distinctly different from controls on important attributes like age group, level of education, number of children alive, type of delivery, tetanus toxoid vaccination during last pregnancy, ever used a contraceptive method. Controls were more numerous in the age group 20–29 years (54.1%) compared to cases (34.7%), but less numerous in the youngest age group 15–19 years (5.9% vs 10.9%) and in older age groups of 30–39 and 40–49 years. More controls had middle or less (23.3%) or secondary and higher education (18.8%) compared to cases (17.7% and 12.9%, respectively). Among cases a higher proportion (19.5%) had seven or more living children compared to controls (11.6%). All controls had tetanus injections (99.9%) compared to cases (86.4%) and more of them ever-used a contraceptive method (36.3%) than cases (11.6%) and higher proportion had a cesarean section (27.9%) compared to the controls (14.4%).

Surprisingly, a higher proportion of cases compared to controls had their last delivery by a skilled birth attendant (83% vs 63.1%) and delivered at a government hospital (43.2% vs 33.4%) while home delivery was more common among controls (32.5%) compared to cases (25.3%). The average level of composite complications index (including high blood pressure, diabetes, anemia, and jaundice during last pregnancy) was similar = 0.62 (standard deviation 0.77) for cases as compared to 0.63 (standard deviation 0.77) for controls. However, cases experienced a higher level (2.35 with standard deviation 1.86) of composite symptoms (fever, fits, vaginal bleeding, jaundice, abdominal pain, breathing difficulty, paleness/anemia, swelling feet or ankles and swelling face during last pregnancy/illness) compared to controls (1.85 with standard deviation 1.62).

### Predictors of maternal mortality

The multivariate analysis controlled for the significant confounding effects of age, education, ever given birth, ever had a cesarean section, had injection in the last pregnancy to prevent tetanus, whether received delivery assistance from a skilled attendant, and ever used a contraceptive method while looking at the conditional odds of each one for maternal death (Table 2). The odds ratios after controlling for factors included in the model were higher for women giving birth at younger age (15–19 years) or at older ages (30–39 and 40–49 years). Age group 20–29 years had the least risk of 0.5 adjusted odds ratio (CI: 0.23–1.07) compared to women in ages 15–19 years. Women with education had lower odds than those with no education—0.35 odds ratio (CI: 0.17–0.74) for women with secondary education and 0.51 (CI: 0.27–0.98) for women with middle or less education—compared to 1.0 for women with no education. Having ever had a cesarean section doubles the odds of maternal death while having had the tetanus injection in the last pregnancy drastically reduced the odds for maternal death close to zero as compared to women who were not vaccinated. Also, having ever-used contraceptive method reduces the odds to 0.21 (CI: 0.11–0.39) compared to 1.0 for never users. Delivery by skilled birth attendant was associated with higher odds of maternal death. Having delivery by a skilled attendant had the matched conditional odds of 4.07 (CI: 2.19–7.57) compared to the reference category of delivery by "others", including home delivery. Women with prior history of symptoms and those who have caesarean section were more likely to have delivery by a skilled attendant and are thus predispose to the risk of death, especially if the quality of care is poor or the attendant lacks the required skills.

**Table 1. Number and percentage distribution (%) of cases and controls by background variables, Pakistan, 2019.**

| Name of Variables | Cases | | Controls | |
|---|---|---|---|---|
| | No. | % | No. | % |
| **Region of residence** | | | | |
| **Punjab** | 32 | 21.77 | 150 | 20.72 |
| **Sindh** | 42 | 28.57 | 210 | 29.01 |
| **KP** | 27 | 18.37 | 134 | 18.51 |
| **Balochistan** | 26 | 17.69 | 130 | 17.96 |
| **GB/ AJ&K** | 20 | 13.61 | 100 | 13.81 |
| **Total** | **147** | **100** | **724** | **100** |
| **Place of residence** | | | | |
| **Urban** | 49 | 33.33 | 242 | 33.43 |
| **Rural** | 98 | 66.67 | 482 | 66.57 |
| **Total** | **147** | **100** | **724** | **100** |
| **Age groups (years)** | | | | |
| **15–19** | 16 | 10.88 | 43 | 5.94 |
| **20–29** | 51 | 34.69 | 392 | 54.14 |
| **30–39** | 66 | 44.9 | 239 | 33.01 |
| **40–49** | 14 | 9.52 | 50 | 6.91 |
| **Total** | **147** | **100** | **724** | **100** |
| **Highest class completed** | | | | |
| **No education** | 102 | 69.39 | 419 | 57.87 |
| **Middle or less** | 26 | 17.69 | 169 | 23.34 |
| **Secondary and higher** | 19 | 12.93 | 136 | 18.78 |
| **Total** | **147** | **100** | **724** | **100** |
| **Ever given birth** | | | | |
| **Yes** | 128 | 87.07 | 698 | 96.41 |
| **No** | 19 | 12.93 | 26 | 3.59 |
| **Total** | **147** | **100** | **724** | **100** |
| **Number of children alive** | | | | |
| **3 and less** | 64 | 50.00 | 446 | 61.6 |
| **Between 4–6** | 39 | 30.47 | 194 | 26.8 |
| **7 and above** | 25 | 19.53 | 84 | 11.6 |
| **Total** | **128** | **100** | **724** | **100** |
| **Ever had a caesarean section operation** | | | | |
| **Yes** | 41 | 27.89 | 104 | 14.36 |
| **No** | 106 | 72.11 | 620 | 85.64 |
| **Total** | **147** | **100** | **724** | **100** |
| **Did see anyone for antenatal care** | | | | |
| **Yes** | 125 | 85.03 | 636 | 87.85 |
| **No** | 22 | 14.97 | 88 | 12.15 |
| **Total** | **147** | **100** | **724** | **100** |
| **Who did she see for antenatal care** | | | | |
| **Did not see anyone** | 22 | 14.97 | 88 | 12.15 |
| **Obstetrician/Specialist** | 59 | 40.14 | 330 | 45.58 |
| **Doctor** | 49 | 33.33 | 232 | 32.04 |
| **Nurse/Midwife/LHV** | 13 | 8.84 | 64 | 8.84 |
| **Nonskilled Birth Attendants** | 4 | 2.72 | 10 | 1.38 |
| **Total** | **147** | **100** | **724** | **100** |

(*Continued*)

**Table 1.** (Continued)

| Name of Variables | Cases | | Controls | |
|---|---|---|---|---|
| | No. | % | No. | % |
| **Trimester of pregnancy at first health provider visit** | | | | |
| **1st trimester** | 63 | 50.4 | 358 | 56.29 |
| **2nd trimester** | 37 | 29.6 | 200 | 31.45 |
| **3rd trimester** | 25 | 20.0 | 78 | 12.26 |
| **Total** | **125** | **100** | **636** | **100** |
| **During last pregnancy had an injection to prevent tetanus** | | | | |
| **Yes** | 127 | 86.39 | 723 | 99.86 |
| **No** | 20 | 13.61 | 1 | 0.14 |
| **Total** | **147** | **100** | **724** | **100** |
| **Place of delivery** | | | | |
| **Home** | 24 | 25.26 | 215 | 32.48 |
| **Govt. hospital/other public** | 41 | 43.16 | 221 | 33.38 |
| **Pvt. hospital/clinic** | 30 | 31.58 | 226 | 34.14 |
| **Total** | **95** | **100** | **662** | **100** |
| **Assistance at delivery** | | | | |
| **Skilled Birth Attendants** | 122 | 82.99 | 457 | 63.12 |
| **Others** | 25 | 17.01 | 267 | 36.88 |
| **Total** | **147** | **100** | **724** | **100** |
| **How was the delivery** | | | | |
| **Normal** | 58 | 61.05 | 328 | 73.54 |
| **assisted vaginal** | 7 | 7.37 | 14 | 3.14 |
| **caesarean section** | 30 | 31.58 | 104 | 23.32 |
| **Total** | **95** | **100** | **446** | **100** |
| **Ever used a Contraceptive method** | | | | |
| **Yes** | 17 | 11.56 | 263 | 36.33 |
| **No** | 130 | 88.44 | 461 | 63.67 |
| **Total** | **147** | **100** | **724** | **100** |

## Discussion

Key findings: We assessed the socio-demographic and clinical factors associated with maternal mortality in Pakistan. We found that cases and controls were similar on access to antenatal care (ANC) and ANC provider but differed on age, education, number of pregnancies, type of delivery, tetanus toxoid vaccination during last pregnancy, and contraceptive use. A higher proportion of cases had deliveries by skilled birth attendants (83%) compared to controls (63%) while home deliveries were more common among controls (32%) compared to cases (25%). Odds of maternal death were lowest among women aged 20–29 years (odds ratio–OR: 0.5; 95% CI 0.23–1.07) and those with secondary or higher education (OR: 0.35; 95% CI 0.17–0.74). Surprisingly, deliveries attended by skilled birth attendants were associated with higher odds of maternal death (OR: 4.07; 95% CI 2.19–7.57) compared to those who were not.

The findings indicate that women in the age group of 20–29 years, who were educated, had ever used a contraceptive method, or had tetanus injection during the last pregnancy had lower odds of maternal death. These findings are expected and consistent with findings from other studies both in Pakistan [7, 8, 11, 12] and elsewhere. However, the finding that women whose last delivery was by a skilled birth attendant had higher odds of maternal death is counterintuitive, though consistent in both PDHS 2006–07 [8] and our analysis of PMMS data.

**Table 2. Multivariate conditional logistic and unconditional logistic regression of predictors of maternal mortality, Pakistan, 2019.**

| Name of Variables | Conditional logistic regression (N = 871) | | | Unconditional logistic regression (N = 871) | | | Goodness of fit |
|---|---|---|---|---|---|---|---|
| | Matched Adjusted OR's | 95% CI's | p value | Adjusted OR's | 95% CI's | p value | |
| Age groups (years) | | | | | | | N = 871 |
| 15–19 | 1 | - | - | 1 | - | - | |
| 20–29 | 0.50 | (0.23–1.07) | 0.075 | 0.47 | (0.21–0.93) | 0.031 | |
| 30–39 | 1.34 | (0.61–2.95) | 0.467 | 1.24 | (0.59–2.61) | 0.564 | |
| 40–49 | 1.21 | (0.42–3.45) | 0.726 | 1.06 | (0.40–2.83) | 0.912 | |
| Highest class completed | | | | | | | |
| No education | 1 | - | - | 1 | - | - | |
| Middle or less | 0.51 | (0.27–0.98) | 0.042 | 0.57 | (0.33–0.99) | 0.046 | |
| Secondary and higher | 0.35 | (0.17–0.74) | 0.006 | 0.49 | (0.27–0.90) | 0.021 | |
| Ever given birth | | | | | | | |
| Yes | 0.19 | (0.09–0.39) | 0.000 | 0.21 | (0.10–0.45) | 0.000 | Hosmer and lemeshow Goodness |
| No | 1 | - | - | 1 | - | - | of fit–test statistics |
| Ever had a caesarean section operation | | | | | | | $(chi^2)$ = 62.39 |
| Yes | 2.05 | (1.18–3.55) | 0.011 | 2.22 | (1.35–3.66) | 0.002 | |
| No | 1 | - | - | 1 | - | - | p value = 0.9380 |
| During last pregnancy had an injection to prevent tetanus | | | | | | | (model is good fit) |
| Yes | 0.007 | (0.001–0.06) | 0.000 | 0.005 | (0.001–0.046) | 0.000 | |
| No | 1 | - | - | 1 | - | - | |
| Assistance at delivery | | | | | | | |
| Skilled Birth Attendants | 4.07 | (2.19–7.57) | 0.000 | 4.13 | (2.32–7.35) | 0.000 | |
| Others | 1 | - | - | 1 | - | - | |
| Ever Used a Contraceptive method | | | | | | | |
| Yes | 0.21 | (0.11–0.39) | 0.000 | 0.22 | (0.12–0.39) | 0.000 | |
| No | 1 | - | - | 1 | - | - | |

*Matching variables in relation to urban–rural status and province of residence were retained in the model as matching variables

There are three explanations for this unexpected result derived from the in-depth analysis of the verbal autopsies of the deceased women identified in the PMMS [9]. First, women after having sought care from diverse types of care providers for symptoms/complications of pregnancy reached the skilled attendant with a severe complication/symptom(s) in a critical condition in many cases after shuffling between two to three facilities before reaching the final referral facility. The in-depth analysis further showed that most maternal deaths occurred at health facilities following delays in deciding to seek professional care and in reaching an appropriate health facility for care [9]. In a social-cultural context with little or no birth planning and accessing care when situation is worsened, this selectivity in terms of women with a complicated pregnancy or delivery reaching late a skilled attendant, manifested in a higher risk of death. Also, women with prior history of symptoms or those requiring cesarean section are more likely to approach skilled attendant for delivery. Such a pattern is noted where an effective referral system is not operational. A study of rural areas of Balochistan and North-West Frontier Province (now renamed as Khyber Pakhtunkhwa) showed a similar pattern of elevated risk of maternal mortality for women who had delivered by a skilled attendant compared to those by a family member or traditional birth attendant [11]. A similar pattern was also observed in eight urban squatter settlements of Karachi [12].

Second, it was found that skilled attendants were not adequately trained and provided a sub-optimal quality of care. The in-depth analysis shows that 36% of all maternal deaths, direct or indirect, and 91% of direct maternal deaths were due to surgical or medical misadventures [9]. Indifference by unmotivated staff, poor skills of health care providers, and lack of medicine and equipment together contributed to heavy death toll of women reaching the facility in reasonable condition [9]. Third, the finding of higher maternal mortality among women with births attended by a skilled provider is consistently reported in studies from Pakistan and the qualitative analysis of verbal autopsies indicates that women with complicated pregnancy or delivery to start with were more likely to seek care from a skilled attendant. The pathway to skilled delivery is, therefore, shaped by the prior history of symptoms exacerbating pregnancy complications leading to institutional care and delivery by a skilled attendant. In addition, the poor quality of service, including poor skills of providers contribute to higher maternal mortality for women with delivery by a skilled attendant. This noteworthy finding indicates that the emphasis on institutional deliveries and delivery by a skilled birth attendant to reduce maternal mortality is insufficient to reduce maternal mortality where norms for pregnancy care are lacking, but a necessary concomitant of complicated pregnancies. Moreover, it is imperative to ensure the quality of care and enhance the skills of health care providers to effectively manage the serious complications of women reaching the facility.

Despite the progress made in reducing maternal mortality, approximately 1 in 143 women in Pakistan will die during her lifetime due to complications during pregnancy, childbirth/abortion, or during the 42 days following pregnancy termination. WHO estimated that 98,000 maternal deaths occurred in Pakistan in 2020 [1]. Substantial inequalities also continue to persist by region, urban-rural place of residence and by subgroups of population in Pakistan. The greatest decline from 2006–07 to 2019 was observed in Balochistan province which nevertheless continues to exhibit the highest MMR of 298 (CI: 130–466) per 100,000 live births in 2019 compared to any other region [3]. Compared to 2006–07, progress was also noted in 2019 for an increase in literacy rate and higher educational attainment, especially in rural areas [9]. Higher order (6 or more) births declined from 22% to 15%; four or more ANC visits nearly doubled from 28% to 52% as well as the visit to obstetrician/gynecologist and doctor for ANC from 33% to 60% [2, 3]. Not having even one ANC visit during the last pregnancy declined from 35% to 8%. The coverage of mobile phones in rural areas also doubled from 46% in 2006–07 to 93% in 2019 [2, 3].

This progress has been, however, slow, and uneven. Women in rural areas, with no education and those living in Balochistan province continue to suffer excessive risk of maternal death than their counterparts in urban areas, living in other regions of Pakistan or educated, especially those attaining higher levels of education. In addition, little progress has been made in rural infrastructure in terms of improved access to the nearest functioning basic health unit (BHU), rural health center (RHC), secondary/tertiary hospital or the availability of motorized public transport [9].

Limitations: Estimating maternal mortality and its risk factors require an exceptionally large sample size that is often not feasible. To keep the sample size within manageable limits, a three-year recall of births and deaths was used in PDHS 2006–07 and PMMS 2019. This approach has problems in that the recall of deaths may have declined during the second and third years before the survey, due to recall errors, misreporting of dates, and/or dissolution or change in the composition of households. It is, therefore, possible that recall errors for the second or third year before the survey underreported maternal deaths. Also, information on the causes and circumstances of deaths ascertained through the verbal autopsy may be less dependable for deaths that occurred in the second or third years of recall. Another potential limitation is intentional or unintentional misreporting. Given the cultural sensitivity, induced

abortions are notoriously underreported or misreported, especially in the restrictive legal contexts such as Pakistan. It is, therefore, possible that some of the induced abortions and related deaths were under-reported.

## Recommendations

High levels of maternal mortality, inequity in the burden of maternal mortality and the finding that women having delivery by a skilled attendant suffer higher risk of death indicate several policy and programmatic implications. First, the health system needs to improve the quality of the obstetric care provided, especially within the public facilities. The government must properly equip facilities and institute proper accountability mechanisms so that all deaths are audited and accounted for. Second, pre-marital counselling should be launched to discourage childbearing during adolescence. Both young (<20 years) and older women of ages over 30 should be given priority attention during ANC, delivery, and the postnatal period. Third, family planning should be promoted for maternal and child survival through birth spacing and prevention of unintended and high-risk pregnancies. According to the PDHS 2017–18, nearly 18% of women became pregnant six to 17 months after a live birth and 37% within 24 months. PDHS data show that the highest proportion of closely spaced pregnancies occur in the adolescent age group of 15–19 years. Compared to older women, this group also has the highest unmet need for birth spacing—one out of six women in this age bracket wants to space their pregnancies but is unable to do so.

Unmet need for family planning (FP) stems from the inability of women to access services. The Lady Health Worker (LHW) program, launched in 1994, had the mandate to provide doorstep FP services to rural women, who have higher unmet need and more unintended pregnancies than urban women. This highly acclaimed program is currently plagued with many issues, the foremost of which is a perennial shortage of contraceptive supplies. Within public health facilities, service providers do not consider offering family planning services and counselling as their responsibility. This needs to be changed through some bold decisions by the government. A major step should be mandatory provision of FP services through the public and private health networks. This has several advantages, including avoiding the hesitancy some women and men have in visiting the socially stigmatized Family Planning centers run by the Population Welfare Department. General health facility visits offer several opportunities for discussing family planning, such as during antenatal care visits, immediately after delivery, during postnatal checkups, and child immunization visits. On these occasions, couples can be encouraged to discuss their family planning needs and individually focused options can be suggested to help meet their needs. This service delivery approach would also allow men to discuss family planning with male health care providers.

The Population Council estimates that even without increasing the current coverage of skilled birth attendance; by simply meeting the 17% unmet need for family planning and thus raising current contraceptive use from 34% to 51%, Pakistan could lower maternal mortality by around 30% and save every year 4,000 maternal lives. With increasing number of women continuing to enter reproductive ages without little or no concomitant rise in modern contraceptive use, women will continue to experience high number of pregnancies that expose them to risks of maternal morbidity and mortality.

Fourth, tetanus toxoid injection during pregnancy should be universal because of its protective effect. Pakistan must strengthen its immunization program and ensure that the injection is made available at both public and private sector facilities. Fifth, a functional and efficient referral system needs to be in place that prevents women shuffling between facilities and instead reach an appropriate facility equipped to provide comprehensive obstetric care. For

ensuring the functional integrity of the referral system, an ambulance system must be put in place that connects the lower to the higher referral facilities. Interventions that make optimal use of the high coverage of mobile phones with home visits by Lady Health Workers and Community Midwives are needed for the continuum of care for maternal, neonatal and child health. The emphasis on institutional delivery and by a skilled birth attendant must be accompanied with strengthening quality of care and providers' skills, especially in public sector hospitals and facilities that are accessed by poor women and those with little or no education. Concerted efforts and investments are urgently needed to meet the public health and human rights imperative of saving maternal lives. This will also enable Pakistan to meet the Sustainable Development Goal 3 target 3.1 to reduce MMR to less than 70 per 100,000 live births by 2030.

## Acknowledgments

We thank the National Institute of Population Studies (NIPS) for providing the PMMS 2019 data and members of the Technical Advisory Committee for helpful comments and suggestions.

## Author Contributions

**Conceptualization:** Ahsan Maqbool Ahmad, Iqbal H. Shah, Ali Muhammad Mir, Muddassir Altaf Bosan.

**Data curation:** Ahsan Maqbool Ahmad, Maqsood Sadiq, Muddassir Altaf Bosan.

**Formal analysis:** Ahsan Maqbool Ahmad, Iqbal H. Shah, Muddassir Altaf Bosan.

**Methodology:** Ahsan Maqbool Ahmad, Iqbal H. Shah, Maqsood Sadiq.

**Supervision:** Ahsan Maqbool Ahmad.

**Validation:** Ahsan Maqbool Ahmad, Maqsood Sadiq.

**Visualization:** Ahsan Maqbool Ahmad, Ali Muhammad Mir, Muddassir Altaf Bosan.

**Writing – original draft:** Ahsan Maqbool Ahmad, Iqbal H. Shah, Ali Muhammad Mir.

**Writing – review & editing:** Ahsan Maqbool Ahmad, Iqbal H. Shah, Ali Muhammad Mir, Muddassir Altaf Bosan.

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
