## [Decision Letter · Decision Letter 0]

4 Mar 2024

PONE-D-23-22204Why mothers continue to die in Pakistan: a nested case-control study of predictors of maternal mortalityPLOS ONE

Dear Dr. Maqbool Ahmad,

Thank you for submitting your manuscript to PLOS ONE. After careful consideration, we feel that it has merit but does not fully meet PLOS ONE’s publication criteria as it currently stands. Therefore, we invite you to submit a revised version of the manuscript that addresses the points raised during the review process.

We look forward to receiving your revised manuscript.

Kind regards,

Abera Mersha, MSc.

Academic Editor

PLOS ONE

Journal Requirements:

Did you know that depositing data in a repository is associated with up to a 25% citation advantage (https://doi.org/10.1371/journal.pone.0230416)? If you’ve not already done so, consider depositing your raw data in a repository to ensure your work is read, appreciated and cited by the largest possible audience. You’ll also earn an Accessible Data icon on your published paper if you deposit your data in any participating repository (https://plos.org/open-science/open-data/#accessible-data).

Reviewers' comments:

Reviewer's Responses to Questions

**Comments to the Author**

1. Is the manuscript technically sound, and do the data support the conclusions?

Reviewer #1: Yes

Reviewer #2: Partly

Reviewer #3: Partly

2. Has the statistical analysis been performed appropriately and rigorously? 

Reviewer #1: Yes

Reviewer #2: I Don't Know

Reviewer #3: Yes

3. Have the authors made all data underlying the findings in their manuscript fully available?

Reviewer #1: Yes

Reviewer #2: Yes

Reviewer #3: Yes

4. Is the manuscript presented in an intelligible fashion and written in standard English?

Reviewer #1: Yes

Reviewer #2: Yes

Reviewer #3: No

5. Review Comments to the Author

Reviewer #1: 1. The study presents the results of original research.

I am wondering this manuscript. I found publish in the medRxiv preprint doi: https://doi.org/10.1101/2023.08.10.23293928; this version posted August 15, 2023.

2. Results reported have not been published elsewhere.

I am wondering this manuscript. I found publish in the medRxiv preprint doi: https://doi.org/10.1101/2023.08.10.23293928; this version posted August 15, 2023.

3. Experiments, statistics, and other analyses are performed to a high technical standard and are described in sufficient detail.

No comment

4. Conclusions are presented in an appropriate fashion and are supported by the data.

No comment

5. The article is presented in an intelligible fashion and is written in standard English.

Yes it is “The article is presented in an intelligible fashion and is written in standard English”

6. The research meets all applicable standards for the ethics of experimentation and research integrity.

The ethics of experimentation and research integrity issue did not show in manuscript. If the project used the secondary data, IRB should be explained about exempt.

7. The article adheres to appropriate reporting guidelines and community standards for data availability.

No comment

Reviewer #2: Why Mothers Continue To Die In Pakistan: A Nested Case-Control Study Of Predictors Of Maternal Mortality

Summary of the Research

Maternal mortality ratio (MMR) declined in Pakistan from 276 maternal deaths per 100,000 live births in 2006-07 to 186 in 2019. Despite this decline, reasons why mothers continue to die, the inequity in the burden of maternal mortality and its predictors largely remain unknown. The in-depth analysis of Pakistan Maternal Mortality Survey 2019 was undertaken using the nested case-control design. Cases and controls were similar on access to antenatal care (ANC) and ANC provider, but differed on age, education, number of pregnancies, type of delivery, tetanus toxoid vaccination during last pregnancy, and ever using a contraceptive method. More of the cases had their last delivery by a skilled birth attendant (83% compared to 63% among controls) and delivered at the government hospital (43% compared to 33% among controls) while home delivery was relatively more common among controls (32% compared to 25% among cases). Odds of maternal death were lowest for women giving birth during age 20-29 (odds ratio – OR: 0.5; 95% CI 0.23-1.07) as compared to women in age 15-19 and those age 30-39 (OR: 1.34; 95% CI 0.61-2.95) and 40-49 (OR: 1.21; 95% CI 0.42-3.45), and among women with secondary or higher education (OR: 0.35; 95% CI 0.17-0.74) compared to women with no education. Women in certain sub-groups confront higher risks of maternal death. Increasing female education, preventing early and late childbearing through contraceptive use, increasing tetanus vaccination during pregnancy, improving providers’ skills, and quality of health care are required to eliminate preventable maternal deaths in Pakistan.

Areas for improvement

Title:

Tile is concise and good well expressing and reflecting the content of the study.

Abstract:

The abstract is well written, however, the authors should revise the language to improve readability.

The authors should make sure that the abstract don’t exceed 300 words.

Introduction:

The authors should write details in introduction section to tell the reader about the general information about the phenomena under study.

The authors should mention the significant of the study and identify the gap in knowledge as well.

The authors should revise introduction section for grammar issues and language to improve readability.

The authors should clear identify the variables of the study.

The authors should explain theoretical framework of the study.

The authors should mention the significance of the study.

The authors should make this section more clear, so readers will understand what message you wanted them to understand.

Overall

Good and brief introduction section.

Material and Methods:

Clear section but The authors should revise the language to improve readability.

The authors should mention why they selected the design of the study clearly.

The authors should briefly explain what is PMMS and provide the full for of this abbreviation. From where they got this questionnaire.

The authors should mention about verbal autopsy (VA) questionnaire, is it different from PMMS questionnaire.

The authors should mention about other ethical consideration for data control.

The authors should briefly explain about Stata version 14.

I am not sure if there are specific criteria to include the participants, mention that please!

Results, discussion and conclusion:

Clear results, discussion and conclusion sections. But improve grammar and language for readability.

References:

The authors should revise all references according to the guidelines provided.

Make sure that all intext referencing reflecting with in the references list.

Reviewer #3: Well done for the reviewers in working with two databases to address an important global health problems. However, there are still some areas that need major revisions. Please see comments below:

Title: I feel like the title is too broad, at first sight I though this paper included demographic, social, economic, cultural factors associated with maternal mortality in Pakistan. I suggest using something more succinct like, "Demographic, Social and Clinical Factors associated with maternal mortality in Pakistan: A nested case control study"

Abstract: The abstract needs a lot of work, see couple suggestions below;

Background: In Parkistan, there has been a decline of maternal mortality by x% between 2006-2019. However, the MMR remains relatively high at xxxxx. This study aims to assess factors associated with MMR in Pakistan.

Methods: We used data from PMMS 2019. We employed a nested case control design. Cases were defined as.......Controls were defined as. We performed a multivariate regression with the following independent variable. The dependent variable is maternal mortality.

Results: Results are currently overwhelming. consider picking 2-3 from table one. And presenting odds rations for statistically significant predictors and surprising findings.

Introduction: First paragraph could be more succinct and shorter focusing mostly on Pakistan. Currently too wordy. Second paragraph needs to be elaborated more. We need more information on the conceptual framework(what informed it? any established theories, key associations from the prior studies mentioned in the past? Do you have a DAG? Rationale and objective of this study needs to be clearly linked.

Figure 1 is nice clear diagram

Methods: Methods section looks generally good. However see comments in abstract.

Did you perform a multistage cluster design or you performed a secondary data analysis from the survey using a nested control study design?

is there room for Flexibility in a statistical analysis plan. Won't this introduce Bias (line 111-114)

Figure 2: I am not an expert in matching cases and controls so I stand to be corrected. What is the rationale of matching cases and controls by clusters? What is the matching ratio 1 case to 5 controls or 1 case to 4 controls. What exactly is happening here? And is it statistically correct?

Results: Improve structure and clarity in results section. Try to use sub-titles. See also comments in abstract. Read how other articles presents such data.

Discussion: Consider formatting it in this way.

1. Key findings, " We assessed the socio-demographic and clinical factors associated with maternal mortality in Pakistan. We found that xxxxxxxxx

2. Similarities or lack of with existing literature on the topic

3. Limitations

4. Recommendations

6. PLOS authors have the option to publish the peer review history of their article (what does this mean?). If published, this will include your full peer review and any attached files.

Reviewer #1: No

Reviewer #2: **Yes: **Zalikha Khamis Al-Marzouqi

Reviewer #3: No

---

## [Author Response · Author response to Decision Letter 0]

22 May 2024

Response to Reviewers

Reviewer #1: 1. The study presents the results of original research.

I am wondering this manuscript. I found publish in the medRxiv preprint doi: https://doi.org/10.1101/2023.08.10.23293928; this version posted August 15, 2023.

2. Results reported have not been published elsewhere.

I am wondering this manuscript. I found publish in the medRxiv preprint doi: https://doi.org/10.1101/2023.08.10.23293928; this version posted August 15, 2023.

Response: Thank you for pointing this out. We initially submitted the manuscript to BMG Global Health that suggested a referral to another journal, but we declined and submitted the manuscript only to PlosOne, after some revisions. We did not submit the manuscript to medRxiv or to any other journal and are surprised to find it on medRxiv as a “preprint” DOI. We have no idea how medRxiv found and placed the manuscript and referred it as “preprint” version when it has not yet been accepted by PlosOne. We confirm that the manuscript has not been published nor submitted to any other journal. Following the reviewer’s comment, we browsed the DOI version and find it different from the revised one.

 3. Experiments, statistics, and other analyses are performed to a high technical standard and are described in sufficient detail.

No comment

4. Conclusions are presented in an appropriate fashion and are supported by the data.

No comment

5. The article is presented in an intelligible fashion and is written in standard English.

Yes it is “The article is presented in an intelligible fashion and is written in standard English”

6. The research meets all applicable standards for the ethics of experimentation and research integrity.

The ethics of experimentation and research integrity issue did not show in manuscript. If the project used the secondary data, IRB should be explained about exempt.

Response: The manuscript used publically available data of Pakistan Maternal Mortality Survey (PMMS). The authors had no role in data collection that was undertaken by National Institute of Population Studies (NIPS) after obtaining the necessary ethical approvals and complying with ethical requirements of informed consent, privacy and confidentiality. We have added information under Ethical Statement. The PMMS data are available for download from: https://dhsprogram.com/methodology/survey/survey-display-552.cfm. 

7. The article adheres to appropriate reporting guidelines and community standards for data availability.

No comment

Reviewer #2: Why Mothers Continue To Die In Pakistan: A Nested Case-Control Study Of Predictors Of Maternal Mortality

Summary of the Research

Maternal mortality ratio (MMR) declined in Pakistan from 276 maternal deaths per 100,000 live births in 2006-07 to 186 in 2019. Despite this decline, reasons why mothers continue to die, the inequity in the burden of maternal mortality and its predictors largely remain unknown. The in-depth analysis of Pakistan Maternal Mortality Survey 2019 was undertaken using the nested case-control design. Cases and controls were similar on access to antenatal care (ANC) and ANC provider, but differed on age, education, number of pregnancies, type of delivery, tetanus toxoid vaccination during last pregnancy, and ever using a contraceptive method. More of the cases had their last delivery by a skilled birth attendant (83% compared to 63% among controls) and delivered at the government hospital (43% compared to 33% among controls) while home delivery was relatively more common among controls (32% compared to 25% among cases). Odds of maternal death were lowest for women giving birth during age 20-29 (odds ratio – OR: 0.5; 95% CI 0.23-1.07) as compared to women in age 15-19 and those age 30-39 (OR: 1.34; 95% CI 0.61-2.95) and 40-49 (OR: 1.21; 95% CI 0.42-3.45), and among women with secondary or higher education (OR: 0.35; 95% CI 0.17-0.74) compared to women with no education. Women in certain sub-groups confront higher risks of maternal death. Increasing female education, preventing early and late childbearing through contraceptive use, increasing tetanus vaccination during pregnancy, improving providers’ skills, and quality of health care are required to eliminate preventable maternal deaths in Pakistan.

Areas for improvement

Title:

Tile is concise and good well expressing and reflecting the content of the study.

Response: Thank you. Reviewer #3 suggested a different title and we have followed the suggestion and changed it to “Demographic, social and health system factors associated with maternal mortality in Pakistan: A nested case-control study”

Abstract:

The abstract is well written, however, the authors should revise the language to improve readability.

The authors should make sure that the abstract don’t exceed 300 words.

Response: Thank you for the suggestion. We have revised the abstract to simplify and improve the readability. The word count is now 263.

Introduction:

The authors should write details in introduction section to tell the reader about the general information about the phenomena under study.

Response: We have further revised the Introduction and summarized the general information available on maternal mortality in Pakistan. 

The authors should mention the significant of the study and identify the gap in knowledge as well.

Response: Thank you for the suggestion. We have added: “However, there remains a dearth of understanding regarding the contextual, health system, social, and biological factors that significantly influence the reduction of preventable maternal deaths in Pakistan. By identifying subgroups of women with high risk of maternal mortality, this study aims to call for concerted efforts focusing on these groups. The study also aims to pinpoint interventions that are needed to reduce preventable maternal deaths in Pakistan. In addition, it highlights the application of case-control methodology to the analysis of maternal mortality data collected in cross-sectional surveys such as Demographic and Health Surveys and Maternal Mortality Surveys.”

The authors should revise introduction section for grammar issues and language to improve readability.

Response: We have revised the text in Introduction to address the comment.

The authors should clear identify the variables of the study.

The authors should explain theoretical framework of the study.

The authors should mention the significance of the study.

The authors should make this section more clear, so readers will understand what message you wanted them to understand.

Response: Thank you for the above suggestions. We have addressed these in our revisions. The first two suggestions have been addressed under Statistical Analysis section where we have added the following: 

“Operationalizing the conceptual framework shown in Figure 1, maternal death was taken as an outcome variable that was dichotomized into whether the death occurred or not (i.e., cases were maternal deaths, and controls were women who were alive and had a live birth). The proximate determinants of the outcome of maternal death are the health seeking behaviors such as seeking antenatal care (ANC); antenatal care provider; month of pregnancy at first ANC visit; receiving tetanus toxoid (TT) injection during pregnancies and comorbidities such as composite index of complications and composite symptoms. Socio-economic variables examined for their potential role in women level risk factors and health-seeking behaviors to ultimately exacerbate or reduce the risk of maternal death are region and urban/rural place of residence and educational level of women. Women level risk factors include age; parity; number of living children, ever had caesarean section; place of delivery; assistance with delivery; type of delivery and contraceptive usage. Together, these variables attenuate or escalate the risk of maternal mortality.”

Suggestion #3 and 4 are addressed in the revision of Introduction as also explained above.

Overall

Good and brief introduction section.

Material and Methods:

Clear section but The authors should revise the language to improve readability. 

Response: We have revised to simplify and improve readability.

The authors should mention why they selected the design of the study clearly.

Response: We have expanded the text in Materials and Methods to provide the requested information as follows:

PMMS and other cross-sectional surveys covering maternal deaths often use verbal autopsy module to ascertain information about the deceased. Case-control design is an appropriate option to study the outcome of maternal mortality where those who died are classified as “cases” and those who survived as “controls”. The purpose of classifying the two groups on outcome, that is, maternal death in this study, is to investigate the underlying factors that distinguish the two groups.

The authors should briefly explain what is PMMS and provide the full for of this abbreviation. 

Response: Information on PMMS is provided under Materials and Methods. The abbreviation of PMMS is defined when first used in the Abstract and in Introduction and also in the text.

From where they got this questionnaire.

Response: The survey was conducted by the National Institute of Population Studies (NIPS) under the auspices of the Demographic and Health Survey Program. The final report of the survey with all details on sampling procedures and the questionnaires as well as data are available at: https://dhsprogram.com/methodology/survey/survey-display-552.cfm. 

The authors should mention about verbal autopsy (VA) questionnaire, is it different from PMMS questionnaire.

Response: PMMS questionnaire included verbal autopsy module. Information is available at: https://dhsprogram.com/methodology/survey/survey-display-552.cfm. 

The authors should mention about other ethical consideration for data control.

Response: We have expanded information under Ethical Statement.

The authors should briefly explain about Stata version 14.

Response: We have added the information.

I am not sure if there are specific criteria to include the participants, mention that please!

Response: The criteria for the participation of respondents in PMMS is described in the final report of PMMS. See https://dhsprogram.com/publications/publication-FR366-Other-Final-Reports.cfm and for the study mentioned under Materials and Methods and shown in Figure 2. 

Results, discussion and conclusion:

Clear results, discussion and conclusion sections. But improve grammar and language for readability.

Response: We have revised to improve readability.

References:

The authors should revise all references according to the guidelines provided.

Make sure that all intext referencing reflecting with in the references list.

Response: Thank you for suggestion, we have revised the references according to the journal referencing guidelines.

Reviewer #3: Well done for the reviewers in working with two databases to address an important global health problems. However, there are still some areas that need major revisions. Please see comments below:

Title: I feel like the title is too broad, at first sight I though this paper included demographic, social, economic, cultural factors associated with maternal mortality in Pakistan. I suggest using something more succinct like, "Demographic, Social and Clinical Factors associated with maternal mortality in Pakistan: A nested case control study"

Response: Thank you for suggesting the alternative title. We have gratefully accepted your suggestion with a minor change from “clinical” to “health system”.

Abstract: The abstract needs a lot of work, see couple suggestions below;

Background: In Parkistan, there has been a decline of maternal mortality by x% between 2006-2019. However, the MMR remains relatively high at xxxxx. This study aims to assess factors associated with MMR in Pakistan.

Methods: We used data from PMMS 2019. We employed a nested case control design. Cases were defined as.......Controls were defined as. We performed a multivariate regression with the following independent variable. The dependent variable is maternal mortality.

Results: Results are currently overwhelming. consider picking 2-3 from table one. And presenting odds rations for statistically significant predictors and surprising findings.

Response: Thank you for the suggestions. We have revised the Abstract incorporating your suggestions.

Introduction: First paragraph could be more succinct and shorter focusing mostly on Pakistan. Currently too wordy. Second paragraph needs to be elaborated more. We need more information on the conceptual framework (what informed it? any established theories, key associations from the prior studies mentioned in the past? Do you have a DAG? Rationale and objective of this study needs to be clearly linked.

Figure 1 is nice clear diagram

Response: Thank you. We have revised the Introduction and have added information on conceptual framework under Statistical Analysis on the linkages among variables identified in the framework.

We had a TAC (Technical Advisory Committee) that reviewed the study design, analysis and the results.

Methods: Methods section looks generally good. However see comments in abstract.

Did you perform a multistage cluster design or you performed a secondary data analysis from the survey using a nested control study design?

Response: We performed secondary data analysis of the survey by defining cases and controls as part of the analysis of data.

is there room for Flexibility in a statistical analysis plan. Won't this introduce Bias (line 111-114)

Response: By “flexibility”, we implied the ability to conduct analysis of “observational” data with subgroups that experience the event, that is, maternal death, and those who did not and to identify factors associated with the outcome. This does not introduce bias. 

Figure 2: I am not an expert in matching cases and controls so I stand to be corrected. What is the rationale of matching cases and controls by clusters? What is the matching ratio 1 case to 5 controls or 1 case to 4 controls. What exactly is happening here? And is it statistically correct?

Response: We matched cases and controls to have both from the same region and place of residence and thus comparable on this aspect. The ratio of 1 to 4 or 1 to 5 is statistically valid. Of course, a ratio of 1 to 1 is better, but maternal death are (fortunately) less than those who gave birth and were alive. The ratio of 1 case to 5 controls provide statistically valid results. 

Results: Improve structure and clarity in results section. Try to use sub-titles. See also comments in abstract. Read how other articles presents such data.

Response: Thank you for the helpful suggestions. We have summarized results under sub-titles of: (1) Profile of cases and controls and risk factors; and (2) predictors of maternal mortality

Discussion: Consider formatting it in this way.

1. Key findings, " We assessed the socio-demographic and clinical factors associated with maternal mortality in Pakistan. We found that xxxxxxxxx

2. Similarities or lack of with existing literature on the topic

3. Limitations

4. Recommendations

Response: Thank you. We have revised Discussions to incorporate suggestions.

6. PLOS authors have the option to publish the peer review history of their article (what does this mean?). If published, this will include your full peer review and any attached files.

Do you want your identity to be public for this peer review? For information about this choice, including consent withdrawal, please see our Privacy Policy.

Reviewer #1: No

Reviewer #2: Yes: Zalikha Khamis Al-Marzouqi

Reviewer #3: No

---

## [Decision Letter · Decision Letter 1]

16 Jun 2024

PONE-D-23-22204R1Demographic, social and health system factors associated with maternal mortality in Pakistan: A nested case-control studyPLOS ONE

Dear Dr. Maqbool Ahmad,

Thank you for submitting your manuscript to PLOS ONE. After careful consideration, we feel that it has merit but does not fully meet PLOS ONE’s publication criteria as it currently stands. Therefore, we invite you to submit a revised version of the manuscript that addresses the points raised during the review process.

We look forward to receiving your revised manuscript.

Kind regards,

Abera Mersha, MSc.

Academic Editor

PLOS ONE

Reviewers' comments:

Reviewer's Responses to Questions

**Comments to the Author**

1. If the authors have adequately addressed your comments raised in a previous round of review and you feel that this manuscript is now acceptable for publication, you may indicate that here to bypass the “Comments to the Author” section, enter your conflict of interest statement in the “Confidential to Editor” section, and submit your "Accept" recommendation.

Reviewer #2: All comments have been addressed

Reviewer #3: All comments have been addressed

2. Is the manuscript technically sound, and do the data support the conclusions?

Reviewer #2: Yes

Reviewer #3: Partly

3. Has the statistical analysis been performed appropriately and rigorously? 

Reviewer #2: Yes

Reviewer #3: Yes

4. Have the authors made all data underlying the findings in their manuscript fully available?

Reviewer #2: Yes

Reviewer #3: No

5. Is the manuscript presented in an intelligible fashion and written in standard English?

Reviewer #2: Yes

Reviewer #3: Yes

6. Review Comments to the Author

Reviewer #2: Well done team, all the best.

The authors addressed all changes

No more comments for them.

I wish them all the best

Reviewer #3: Abstract

This conclusion is off – how do you make such a conclusion from these results that are only looking at association and not causality?

I would suggest_

“Conclusion: This study provides factors associated with maternal mortality. Further investigation is needed on the determinants positively associated with high mortality rates.”

There is a limitation on what you can say when assessing causality or maybe you can argue that maternal deaths are rare and hence odds ratio = relative risk ratio, read up on it.

Introduction

Improved – However I would remove sections that speaks about the methodology in the introduction

There is a specific section for methodology

Methodology

Please follow the following format for clarity:

Design: Case-control design of PMMS and Surveys with Maternal Mortality data (Include years)

Setting and Population

Data Collection

Study Variable – including Primary Outcome

Type of analysis and Software

Ethical Considerations

Figure 2: Scheme of cases an 132 d controls identification and matching is a great diagram, give the explanation you gave on the rationale of different matching ratios

Results

“Multi-stage analyses included the phases of descriptive analysis and inferential analyses by applying

173 univariate and multivariate regression techniques. The proportional representation of cases versus controls

174 was similar for the matching variables of region/province of residence and place of residence (i.e. urban vs

175 rural).” - Belongs in the methodology section.

Begin like – Table shows the Number and percentage distribution (%) of cases and controls by background variables, Pakistan, 2019…………

Discussion

The discussion needs restructuring.

We assessed the socio-demographic and clinical factors associated with maternal mortality in Pakistan. We found that We Cases and controls were similar on access to antenatal care (ANC) and ANC provider but differed on age, education, number of pregnancies, type of delivery, tetanus toxoid vaccination during last pregnancy, and contraceptive usage. A higher proportion of cases had deliveries by skilled birth attendants (83% compared to 63% among controls) while home deliveries were more common among controls (32% compared to 25% among cases). Odds of maternal death were lowest among women aged 20-29 years (odds ratio – OR: 0.5; 95% CI 0.23-1.07) and those with secondary or higher education (OR: 0.35; 95% CI 0.17-0.74). Surprisingly, deliveries attended by skilled birth attendants were associated with higher odds of maternal death (OR: 4.07; 95% CI 2.19-7.57) compared to those who were not.

Second paragraph should speak of similarities or lack of with existing literature on the topic

Third paragraph should speak of of limitations which you already have.

Fourth paragraph should speak of Recommendations (please see comments in abstract.

7. PLOS authors have the option to publish the peer review history of their article (what does this mean?). If published, this will include your full peer review and any attached files.

Reviewer #2: **Yes: **Zalikha Khamis Al-Marzouqi

Reviewer #3: No

---

## [Author Response · Author response to Decision Letter 1]

31 Jul 2024

Reviewers' comments:

Reviewer's Responses to Questions

Comments to the Author

1. If the authors have adequately addressed your comments raised in a previous round of review and you feel that this manuscript is now acceptable for publication, you may indicate that here to bypass the “Comments to the Author” section, enter your conflict of interest statement in the “Confidential to Editor” section, and submit your "Accept" recommendation.

Reviewer #2: All comments have been addressed

Reviewer #3: All comments have been addressed

2. Is the manuscript technically sound, and do the data support the conclusions?

Reviewer #2: Yes

Reviewer #3: Partly

Response: We have revised the text in line with the comments and suggestions received.

3. Has the statistical analysis been performed appropriately and rigorously? 

Reviewer #2: Yes

Reviewer #3: Yes

4. Have the authors made all data underlying the findings in their manuscript fully available?

Reviewer #2: Yes

Reviewer #3: No

Response: Pakistan Maternal Mortality Survey 2019 data are in public domain and can be downloaded from the website: https://dhsprogram.com/data/available-datasets.cfm. The survey was conducted by the National Institute of Population Studies. 

5. Is the manuscript presented in an intelligible fashion and written in standard English?

Reviewer #2: Yes

Reviewer #3: Yes

6. Review Comments to the Author

Reviewer #2: Well done team, all the best.

The authors addressed all changes

No more comments for them.

I wish them all the best

Reviewer #3: Abstract

This conclusion is off – how do you make such a conclusion from these results that are only looking at association and not causality?

I would suggest_

“Conclusion: This study provides factors associated with maternal mortality. Further investigation is needed on the determinants positively associated with high mortality rates.”

There is a limitation on what you can say when assessing causality or maybe you can argue that maternal deaths are rare and hence odds ratio = relative risk ratio, read up on it.

Response: Thank you. We agree with the comment and suggestion and have revised the text to: “This study identifies factors associated with maternal mortality in Pakistan. Further investigation is needed on the determinants of high maternal mortality.”

Introduction

Improved – However I would remove sections that speaks about the methodology in the introduction

There is a specific section for methodology

Response: Thank you for the helpful suggestion. We have moved the relevant text to Methods section. 

Methodology

Please follow the following format for clarity:

Design: Case-control design of PMMS and Surveys with Maternal Mortality data (Include years)

Setting and Population

Data Collection

Study Variable – including Primary Outcome

Type of analysis and Software

Ethical Considerations

Response: We have followed the suggested format and reorganized the text into the suggested sub-sections.

Figure 2: Scheme of cases and controls identification and matching is a great diagram, give the explanation you gave on the rationale of different matching ratios

Response: Added in the text of manuscript as follows (row 145-149)

There were a total of 147 maternal deaths (eligible cases) in the PMMS 2019 survey. To maximize the statistical power of analysis, inclusivity for statistical analysis purposes was considered in a way, so that the information on maximum number of cases could be utilized. In this milieu, there were 136 cases were included in the final analysis on the basis that they had at-least one cluster matched control available. As a result, there were only four clusters for whom lesser number of controls per case (i.e. less than ratio of 1:5) were available. 

Results

“Multi-stage analyses included the phases of descriptive analysis and inferential analyses by applying

173 univariate and multivariate regression techniques. The proportional representation of cases versus controls

174 was similar for the matching variables of region/province of residence and place of residence (i.e. urban vs

175 rural).” - Belongs in the methodology section.

Begin like – Table shows the Number and percentage distribution (%) of cases and controls by background variables, Pakistan, 2019…………

Response: Thank you for the helpful suggestions. We have moved the text, as suggested to Methodology section. 

Discussion

The discussion needs restructuring.

We assessed the socio-demographic and clinical factors associated with maternal mortality in Pakistan. We found that We Cases and controls were similar on access to antenatal care (ANC) and ANC provider but differed on age, education, number of pregnancies, type of delivery, tetanus toxoid vaccination during last pregnancy, and contraceptive usage. A higher proportion of cases had deliveries by skilled birth attendants (83% compared to 63% among controls) while home deliveries were more common among controls (32% compared to 25% among cases). Odds of maternal death were lowest among women aged 20-29 years (odds ratio – OR: 0.5; 95% CI 0.23-1.07) and those with secondary or higher education (OR: 0.35; 95% CI 0.17-0.74). Surprisingly, deliveries attended by skilled birth attendants were associated with higher odds of maternal death (OR: 4.07; 95% CI 2.19-7.57) compared to those who were not.

Second paragraph should speak of similarities or lack of with existing literature on the topic

Third paragraph should speak of of limitations which you already have.

Fourth paragraph should speak of Recommendations (please see comments in abstract.

Response: We have restructured the Discussion and followed these helpful recommendations. 

7. PLOS authors have the option to publish the peer review history of their article (what does this mean?). If published, this will include your full peer review and any attached files.

Do you want your identity to be public for this peer review? For information about this choice, including consent withdrawal, please see our Privacy Policy.

Reviewer #2: Yes: Zalikha Khamis Al-Marzouqi

Reviewer #3: No

---

## [Decision Letter · Decision Letter 2]

8 Oct 2024

PONE-D-23-22204R2Demographic, social and health system factors associated with maternal mortality in Pakistan: A nested case-control studyPLOS ONE

Dear Dr. Maqbool Ahmad,

Thank you for submitting your manuscript to PLOS ONE. After careful consideration, we feel that it has merit but does not fully meet PLOS ONE’s publication criteria as it currently stands. Therefore, we invite you to submit a revised version of the manuscript that addresses the points raised during the review process.

Thank you for addressing the initial comments provided by the reviewers. Based on the revised manuscript, the reviewers have requested that some minor comments still need to be addressed. 

We look forward to receiving your revised manuscript.

Kind regards,

Muhammad Haroon Stanikzai

Academic Editor

PLOS ONE

Journal Requirements:

Reviewers' comments:

Reviewer's Responses to Questions

**Comments to the Author**

1. If the authors have adequately addressed your comments raised in a previous round of review and you feel that this manuscript is now acceptable for publication, you may indicate that here to bypass the “Comments to the Author” section, enter your conflict of interest statement in the “Confidential to Editor” section, and submit your "Accept" recommendation.

Reviewer #2: All comments have been addressed

Reviewer #4: (No Response)

2. Is the manuscript technically sound, and do the data support the conclusions?

Reviewer #2: Yes

Reviewer #4: Yes

3. Has the statistical analysis been performed appropriately and rigorously? 

Reviewer #2: Yes

Reviewer #4: (No Response)

4. Have the authors made all data underlying the findings in their manuscript fully available?

Reviewer #2: Yes

Reviewer #4: Yes

5. Is the manuscript presented in an intelligible fashion and written in standard English?

Reviewer #2: Yes

Reviewer #4: Yes

6. Review Comments to the Author

Reviewer #2: All Comments had been addressed.

All the best

Reviewer #4: Dear Authors,

It is good Job. The study used nested case-control design with robust statistical analysis to better understand maternal mortality in Pakistan. I have some comments that can improve your work.

- what does your "main findings mean" merely should be stated in abstract section of the conclusion.

- The study shows sigificant association between SBAs and high MMR. Could this works for causality too? or How can the authors address limitations of drawing conclusions on the causality?

- Why pakistan still have high number of MMR despite considerable improvements.

- I see some of the section of the references are not full. Please re consider it

7. PLOS authors have the option to publish the peer review history of their article (what does this mean?). If published, this will include your full peer review and any attached files.

Reviewer #2: **Yes: **Dr. Zalikha Khamis Al-Marzouqi

Reviewer #4: **Yes: **Eskinder Israel

---

## [Author Response · Author response to Decision Letter 2]

31 Oct 2024

Note: Responses are show in Italic.

Journal Requirements:

Please review your reference list to ensure that it is complete and correct. If you have cited papers that have been retracted, please include the rationale for doing so in the manuscript text or remove these references and replace them with relevant current references. Any changes to the reference list should be mentioned in the rebuttal letter that accompanies your revised manuscript. If you need to cite a retracted article, indicate the article’s retracted status in the References list and also include a citation and full reference for the retraction notice.

Response: We have reviewed reference list for completeness and correctness. Changes have been made to correct the title or other information. No retracted paper was (and is) in the reference list. 

Reviewers' comments:

Reviewer's Responses to Questions

Comments to the Author

1. If the authors have adequately addressed your comments raised in a previous round of review and you feel that this manuscript is now acceptable for publication, you may indicate that here to bypass the “Comments to the Author” section, enter your conflict of interest statement in the “Confidential to Editor” section, and submit your "Accept" recommendation.

Reviewer #2: All comments have been addressed

Reviewer #4: (No Response)

Response: Thank you.

2. Is the manuscript technically sound, and do the data support the conclusions?

Reviewer #2: Yes

Reviewer #4: Yes

Response: Thank you.

3. Has the statistical analysis been performed appropriately and rigorously? 

Reviewer #2: Yes

Reviewer #4: (No Response)

Response: Thank you.

4. Have the authors made all data underlying the findings in their manuscript fully available?

Reviewer #2: Yes

Reviewer #4: Yes

Response: Thank you.

5. Is the manuscript presented in an intelligible fashion and written in standard English?

Reviewer #2: Yes

Reviewer #4: Yes

Response: Thank you.

6. Review Comments to the Author

Reviewer #2: All Comments had been addressed.

Response: Thank you.

All the best

Reviewer #4: Dear Authors,

It is good Job. The study used nested case-control design with robust statistical analysis to better understand maternal mortality in Pakistan. I have some comments that can improve your work.

Response: Thank you.

- what does your "main findings mean" merely should be stated in abstract section of the conclusion.

Response: Thank you. We have now included the main findings under Conclusions in the Abstract. Specifically, we have added: “This study identifies secondary or higher maternal education, having had tetanus injection during the last pregnancy, ever-used contraception or being in the age group of 20-29 years were factors associated with lower risk of maternal mortality. Conversely, skilled birth attendance increases the risk of maternal death in Pakistan”. 

- The study shows sigificant association between SBAs and high MMR. Could this works for causality too? or How can the authors address limitations of drawing conclusions on the causality?

Response: Thank you for raising an important point. Out data and statistical technique of case-control analysis do not permit drawing causal relationship between any predictor, including delivery by skil bi9rth attendants. While not drawing causality, we discuss the relationship between delivery by skill birth attendant and risk of maternal death. 

Lines 266-288 of the manuscript discuss the association between SBAs and risk of maternal mortality as follows: “However, the finding that women whose last delivery was by a skilled birth attendant had higher odds of maternal death is counterintuitive, though consistent in both PDHS 2006-07 [8] and our analysis of PMMS data. There are three explanations for this unexpected result derived from the in-depth analysis of the verbal autopsies of the deceased women identified in the PMMS [9]. First, women after having sought care from diverse types of care providers for symptoms/complications of pregnancy reached the skilled attendant with a severe complication/symptom(s) in a critical condition in many cases after shuffling between two to three facilities before reaching the final referral facility. The in-depth analysis further showed that most maternal deaths occurred at health facilities following delays in deciding to seek professional care and in reaching an appropriate health facility for care [9].In a social-cultural context with little or no birth planning and accessing care when situation is worsened, this selectivity in terms of women with a complicated pregnancy or delivery reaching late a skilled attendant, manifested in a higher risk of death. Also, women with prior history of symptoms or those requiring cesarean section are more likely to approach skilled attendant for delivery. Such a pattern is noted where an effective referral system is not operational. A study of rural areas of Balochistan and North-West Frontier Province (now renamed as Khyber Pakhtunkhwa) showed a similar pattern of elevated risk of maternal mortality for women who had delivered by a skilled attendant compared to those by a family member or traditional birth attendant [11]. A similar pattern was also observed in eight urban squatter settlements of Karachi [12]. 

Second, it was found that skilled attendants were not adequately trained and provided a sub-optimal quality of care. The in-depth analysis shows that 36% of all maternal deaths, direct or indirect, and 91% of direct maternal deaths were due to surgical or medical misadventures [9]. Indifference by unmotivated staff, poor skills of health care providers, and lack of medicine and equipment together contributed to heavy death toll of women reaching the facility in reasonable condition [9]. Third, the finding of higher maternal mortality among women with births attended by a skilled provider is consistently reported in studies from Pakistan and the qualitative analysis of verbal autopsies indicates that women with complicated pregnancy or delivery to start with were more likely to seek care from a skilled attendant. The pathway to skilled delivery is, therefore, shaped by the prior history of symptoms exacerbating pregnancy complications leading to institutional care and delivery by a skilled attendant. In addition, the poor quality of service, including poor skills of providers contribute to higher maternal mortality for women with delivery by a skilled attendant. This noteworthy finding indicates that the emphasis on institutional deliveries and delivery by a skilled birth attendant to reduce maternal mortality is insufficient to reduce maternal mortality where norms for pregnancy care are lacking, but a necessary concomitant of complicated pregnancies. Moreover, it is imperative to ensure the quality of care and enhance the skills of health care providers to effectively manage the serious complications of women reaching the facility.”

We discuss limitations of the study in lines 266-288 of the manuscript.

- Why pakistan still have high number of MMR despite considerable improvements.

Response: Thank you for the thought-provoking question. Despite considerable improvement, Pakistan continues to experience high level of maternal mortality as compared to other countries in South Asia. One of the main reason is the growing number of women entering reproductive age while contraceptive use has been stagnated and high number of pregnancies continue to occur many of them unintended. Pregnancies of young women and unintended pregnancies are of high risk of morbidity and mortality. We discuss this aspect in Lines 336-361 with additional text in Lines 359-361 as follows: “Third, family planning should be promoted for maternal and child survival through birth spacing and prevention of unintended and high-risk pregnancies. According to the PDHS 2017-18, nearly 18% of women became pregnant six to 17 months after a live birth and 37% within 24 months. PDHS data show that the highest proportion of closely spaced pregnancies occur in the adolescent age group of 15-19 years. Compared to older women, this group also has the highest unmet need for birth spacing—one out of six women in this age bracket wants to space their pregnancies but is unable to do so. 

Unmet need for family planning (FP) stems from the inability of women to access services. The Lady Health Worker (LHW) program, launched in 1994, had the mandate to provide doorstep FP services to rural women, who have higher unmet need and more unintended pregnancies than urban women. This highly acclaimed program is currently plagued with many issues, the foremost of which is a perennial shortage of contraceptive supplies. Within public health facilities, service providers do not consider offering family planning services and counselling as their responsibility. This needs to be changed through some bold decisions by the government. A major step should be mandatory provision of FP services through the public and private health networks. This has several advantages, including avoiding the hesitancy some women and men have in visiting the socially stigmatized Family Planning centers run by the Population Welfare Department. General health facility visits offer several opportunities for discussing family planning, such as during antenatal care visits, immediately after delivery, during postnatal checkups, and child immunization visits. On these occasions, couples can be encouraged to discuss their family planning needs and individually focused options can be suggested to help meet their needs. This service delivery approach would also allow men to discuss family planning with male health care providers.

The Population Council estimates that even without increasing the current coverage of skilled birth attendance; by simply meeting the 17% unmet need for family planning and thus raising current contraceptive use from 34% to 51%, Pakistan could lower maternal mortality by around 30% and save every year 4,000 maternal lives. With increasing number of women continuing to enter reproductive ages without little or no concomitant rise in modern contraceptive use, women will continue to experience high number of pregnancies that expose them to risks of maternal morbidity and mortality”.

- I see some of the section of the references are not full. Please re consider it

Response: Thank you for the comment. We have reviewed and checked references for completeness and correctness.

7. PLOS authors have the option to publish the peer review history of their article (what does this mean?). If published, this will include your full peer review and any attached files.

Do you want your identity to be public for this peer review? For information about this choice, including consent withdrawal, please see our Privacy Policy.

Reviewer #2: Yes: Dr. Zalikha Khamis Al-Marzouqi

Reviewer #4: Yes: Eskinder Israel

---

## [Decision Letter · Decision Letter 3]

14 Nov 2024

Demographic, social and health system factors associated with maternal mortality in Pakistan: A nested case-control study

PONE-D-23-22204R3

Dear Dr. Maqbool Ahmad,

We’re pleased to inform you that your manuscript has been judged scientifically suitable for publication and will be formally accepted for publication once it meets all outstanding technical requirements.

Kind regards,

Muhammad Haroon Stanikzai

Academic Editor

PLOS ONE

Additional Editor Comments (optional):

Thank you for submitting this important manuscript to PLOS ONE Journal. I wish you great success in your continued efforts to improve care for women in Pakistan.

Reviewers' comments:

Reviewer's Responses to Questions

**Comments to the Author**

1. If the authors have adequately addressed your comments raised in a previous round of review and you feel that this manuscript is now acceptable for publication, you may indicate that here to bypass the “Comments to the Author” section, enter your conflict of interest statement in the “Confidential to Editor” section, and submit your "Accept" recommendation.

Reviewer #4: All comments have been addressed

2. Is the manuscript technically sound, and do the data support the conclusions?

Reviewer #4: Yes

3. Has the statistical analysis been performed appropriately and rigorously? 

Reviewer #4: Yes

4. Have the authors made all data underlying the findings in their manuscript fully available?

Reviewer #4: Yes

5. Is the manuscript presented in an intelligible fashion and written in standard English?

Reviewer #4: Yes

6. Review Comments to the Author

Reviewer #4: Dear Authors,

Thank you very much for your good work and I wish you all the best in your future life carrier.

7. PLOS authors have the option to publish the peer review history of their article (what does this mean?). If published, this will include your full peer review and any attached files.

Reviewer #4: **Yes: **Eskinder Israel

---

## [Editor Report · Acceptance letter]

18 Dec 2024

PONE-D-23-22204R3 

PLOS ONE

Dear Dr. Maqbool Ahmad, 

I'm pleased to inform you that your manuscript has been deemed suitable for publication in PLOS ONE. Congratulations! Your manuscript is now being handed over to our production team.

Kind regards, 

on behalf of

Dr. Muhammad Haroon Stanikzai 

Academic Editor

PLOS ONE